# Evolutionary Trajectories and Genomic Divergence in Localized Breast Cancers after Ipsilateral Breast Tumor Recurrence

**DOI:** 10.3390/cancers13081821

**Published:** 2021-04-11

**Authors:** Chia-Hsin Wu, Hsien-Tang Yeh, Chia-Shan Hsieh, Chi-Cheng Huang, Amrita Chattopadhyay, Yuan-Chiang Chung, Shih-Hsin Tu, Yung-Hua Li, Tzu-Pin Lu, Liang-Chuan Lai, Ming-Feng Hou, King-Jen Chang, Mong-Hsun Tsai, Eric Y. Chuang

**Affiliations:** 1Graduate Institute of Biomedical Electronics and Bioinformatics, National Taiwan University, Taipei 10617, Taiwan; d05945009@ntu.edu.tw (C.-H.W.); yunghua@ntu.edu.tw (Y.-H.L.); 2Department of Surgery, Lotung Poh-Ai Hospital, Yilan County 26546, Taiwan; 836023@mail.pohai.org.tw; 3Genome and Systems Biology Degree Program, National Taiwan University, Taipei 10617, Taiwan; d00b48014@ntu.edu.tw; 4Comprehensive Breast Health Center, Taipei Veterans General Hospital, Taipei 11217, Taiwan; cchuang29@vghtpe.gov.tw; 5Bioinformatics and Biostatistics Core, Centers of Genomic and Precision Medicine, National Taiwan University, Taipei 10055, Taiwan; amrita@ntu.edu.tw; 6Department of Breast Surgery, Dajia Branch, Kuang Tien General Hospital, Taichung 43761, Taiwan; chung11753@ktgh.com.tw; 7Department of Surgery, School of Medicine, College of Medicine, Taipei Medical University, Taipei 11031, Taiwan; drtu@h.tmu.edu.tw; 8Department of Public Health, National Taiwan University, Taipei 10055, Taiwan; tplu@ntu.edu.tw; 9Graduate Institute of Physiology, College of Medicine, National Taiwan University, Taipei 10051, Taiwan; llai@ntu.edu.tw; 10Division of Breast Surgery, Department of Surgery, Kaohsiung Medical University, Kaohsiung 80708, Taiwan; mifeho@kmu.edu.tw; 11Department of Surgery, National Taiwan University Hospital, Taipei 10016, Taiwan; kingjen@ntu.edu.tw; 12Institute of Biotechnology, National Taiwan University, Taipei 10672, Taiwan; 13Biomedical Technology and Device Research Laboratories, Industrial Technology Research Institute, Hsinchu 31057, Taiwan

**Keywords:** ipsilateral breast tumor relapse, whole-exome sequencing, clonal architecture, alteration, actionability

## Abstract

**Simple Summary:**

Ipsilateral breast tumor relapse (IBTR) occurs in 5–10% of localized breast cancers (BCs) within 10 years of incidence, despite proper treatment of the primary lesion. However, the clinical consequences of evolutionary trajectories of BC cells and their impact on IBTR remain poorly understood. Here, we conducted a longitudinal genomic analysis of 10 matched localized BC patients with IBTR. Overall, we identified the differences in homologous recombination deficiency, chromosomal instability, and somatic mutation drivers between primary and relapsed lesions. Our analyses highlighted three clonal architectures that shape by distinct mutagenic processes and subclonal diversification during relapse progression. Finally, this study provided a framework, which integrated actionable biomarkers with clonal architectures, towards improvement of future treatment decisions.

**Abstract:**

The evolutionary trajectories that drive clinical and therapeutic consequences in localized breast cancers (BCs) with ipsilateral breast tumor relapse (IBTR) remain largely unknown. Analyses of longitudinal paired whole-exome sequencing data from 10 localized BC patients with IBTR reveal that, compared to primary breast tumors, homologous recombination (HR) deficiency, inactivation of the HR pathway, chromosomal instability, and somatic driver mutations are more frequent. Furthermore, three major models of evolution in IBTR are summarized, through which relative contributions of mutational signatures shift, and the subclonal diversity expansions are shown. Optimal treatment regimens are suggested by the clinically relevant molecular features, such as HR deficiency (20%) or specific alterations (30%) with sensitivity to available FDA-approved drugs. Finally, a rationale for the development of the therapeutic management framework is provided. This study sheds light on the complicated evolution patterns in IBTR and has significant clinical implications for future improvement of treatment decisions.

## 1. Introduction

Breast cancer (BC) remains the most common solid malignancy in women [1]. Extensive genomic heterogeneity of primary breast tumors is achieved via a sequential series of somatic events through a long mutational history, albeit a curable one when localized (node-negative) [1,2]. However, 5–10% of localized BCs result in ipsilateral breast tumor relapse (IBTR), a relapsed tumor/lesion occurring at the ipsilateral breast following the initial surgery, within 10 years of incidence, despite proper treatment of the primary lesion [3,4]. This localized failure is associated with an elevated risk for the development of incurable distant disease (metastasis) and death [3,5]. As advanced BCs reveal wide genetic divergence among individuals [6], it reinforces the need for a comprehensive genomic analysis of IBTR in localized BC patients.

Previous studies have typically focused on breast tumor characterization for primary lesions [7,8,9,10,11], uncovering a heterogeneous portrait of somatic drivers, underlying mutagenic processes, and common genomic features, including homologous recombination deficiency (HRD). Available knowledge about the genomic repertoire of IBTR is limited to some preliminary observations from single tumor samples taken at one time point in the disease course, which suggest that subclonal stratification of alterations could improve the efficacy of targeted therapies [12]. This strategy, while promising, nonetheless has limitations owing to the underestimation of the true extent of clonal expansion within tumors.

Due to therapeutic pressure and its effect on disease progression, recent studies have begun to provide evidence of tumor evolution in advanced BC [6,13,14,15,16]. These findings confirm the clonal relatedness between metastases and the respective primary lesions; however, the difference in acquisition of additional driver genes in advanced BCs is attributed primarily to the SWI–SNF complex and inactivated JAK2-STAT3 pathways [6]. Therefore, the clinical consequences of evolutionary trajectories of BC cells and their impact on IBTR remain poorly understood.

To address these issues, we conducted a detailed genomic analysis of 10 matched localized BC patients with IBTR that provided valuable insights into (1) the extent of HRD during relapse progression, (2) the bi-allelic inactivation of HR genes involved in tumor maintenance, (3) the divergence of somatic drivers, (4) the changes in mutational stresses over time, (5) the slight clonal relatedness and extensive subclonal diversity among evolution models, and (6) additional potential therapeutic targets (Figure 1). Since the clonal architecture leading to IBTR showed clear clinical utility, we further provided a comprehensive framework with clinically relevant molecular features for determining the benefits of intensification/de-intensification of treatment for curable cancers.

## 2. Results

### 2.1. Overview of Patient Cohorts

Longitudinal high-quality whole-exome sequencing (WES) was performed, at an average depth of 200X, on DNA from 10 localized BC patients with IBTR subsequent to definitive treatment for their primary tumor. For each patient, two longitudinal tumor samples, henceforth termed primary and relapse, were analyzed with matched germline DNA samples. Overall, primary cancers were diagnosed between 27 and 72 years (median 47 years) of age, all at early stages (stage I–II) and most with an intermediate histologic grade (77.8%) (Table 1; Appendix A). Relapse samples were obtained 6–107 months after the primary tumor diagnosis. Half of the patients (50.0%) were classified as having the luminal A subtype (defined as hormone receptor-positive (HR+)/human epidermal growth factor receptor-negative (HER2−)), and the remaining cases were distributed equally between the luminal B (HR+/HER2+; 10.0%), HER2-enriched (HR−/HER2+; 10.0%), and triple-negative (TNBC; HR−/HER2−; 10.0%) subtypes (Table 1; Appendix A). All patients received standard of care treatment following hospital guidelines, such as curative surgery with local radiotherapy, tamoxifen for hormone therapy in HR+ cohorts, herceptin as targeted therapy for HER2+ patients, and cytotoxic chemotherapeutic regimens for consolidation. Treatment histories are summarized in Appendix A.

### 2.2. HRD and Chromosomal Instability (CIN) during BC Evolution

We sought to define whether relapsed tumors would exhibit higher extents in defects of HR DNA repair or harbor more unstable genomes. To evaluate the extent of HRD in primary tumors leading up to relapse, we quantified HRD in tumors by combining three independent genomic scores [17]: (1) HRD-loss of heterozygosity (HRD-LOH) score; (2) the telomeric allelic imbalance (tAI); and (3) the large-scale state transitions (LST) score. Relapsed tumors exhibited a significantly higher extent of HRD compared to primary tumors (*p* = 0.022; one-tailed paired Mann–Whitney U test; Figure 2A). Notably, three relapsed tumors harbored the HRD phenotype (HRD score ≥42 or/and bi-allelic alterations in *BRCA*1/2) [17]. Two cases had an increase in HRD (going from proficient to deficient based on the scores). The so-called *BRCA2*-deficient case was based on bi-allelic alterations without showing a high HRD score.

As HRD was previously identified as the key genomic feature resulting in CIN [18], we hypothesized that tumors with a higher extent of HRD exhibit a higher level of CIN in our study cohort. We quantified CIN as the fraction of the autosomal chromosomes with a non-diploid copy number, for each tumor, and examined if the level of CIN was associated with the extent of HRD. Consistent with our hypothesis, we observed a strong positive correlation between CIN and HRD across the whole cohort (Pearson correlation coefficient 0.82, *p* = 9.625 × 10^−6^). However, in relapsed tumors, the level of CIN was apparently higher compared with matched primary tumors in only two out of 10 patients, and another four cases had a slightly increased level of CIN, which was not statistically significant. (*p* = 0.116; one-tailed Mann–Whitney U test; Figure 2B). There was a trend toward higher HRD scores among BC patients with increased CIN, with one major exception (BC10) due to the whole-genome doubling (WGD). Furthermore, the two relapsed tumors (BC02 and BC05) with the highest HRD scores also harbored the highest levels of CIN. This confirms that the HRD phenotype is frequently accompanied by CIN in relapsed tumors.

Next, we sought to investigate the presence of shared bi-allelic and monoallelic alterations affecting HR-related genes that might determine their involvement in tumor maintenance from primary tumors to their relapse. In total, three out of 10 tumors (BC05, BC07, and BC10) harbored a shared bi-allelic alteration in the HR gene (30%; Figure 2C), compared to the TCGA cohort, with 10% of BCs having such alterations [19]. Among these three tumors, there exist germline loss-of-function (LOF) mutations with somatic LOH in genes such as *ATM*, *MRE11*, *FANCM*, *FANCI*, *WRN*, and *BRCA2*. Notably, the increase in HRD in patient BC05 (Figure 2A) might be driven by the bi-allelic inactivation of *WRN*, where a mono-allelic germline alteration in the primary tumor progresses to the bi-allelic alteration by harboring a “second-hit” [20], somatic LOH in the relapsed tumor (see change from red to blue in *WRN*). Moreover, the total number of bi-allelic and mono-allelic alterations in HR-related genes for most tumors suggests that a higher number of alterations might contribute to the increased HRD seen in relapsed tumors.

### 2.3. Somatic Drivers of Relapse

On average, the relapsed tumors harbored 57% more mutations than the primary tumors, notwithstanding considerable variability among patients. The additional mutation burden in the relapsed tumors suggests that the rate at which mutations accumulate was generally higher in relapse tumors than in primary tumors during tumorigenesis (Figure 3). 

To investigate the landscape of driver mutations from primary tumor until relapse, we manually curated the BC driver genes known to contain single nucleotide variations (SNVs) and insertions/deletions (INDELs) as reported in previous studies [6,8,21,22,23]. A total of 11 and 13 cancer genes were identified as drivers in the primary and relapsed samples, respectively, of which six genes were present in both samples (Figure 3). Mutations in well-known BC genes, such as *PIK3CA* and *CTCF*, were found in both primary and relapsed lesions, along with *GATA3*, which was more recurrently present in these lesions. Notably, mutations in *CTCF* and *ERBB2* were hotspot sites of BC, as has previously been associated with endocrine resistance [24]. PI3K signaling (*PIK3CA*, *AKT1*, and *PTEN*) was found to be frequently altered compared to other pathways in our cohort, followed by receptor-tyrosine kinase-Ras (RTK-RAS; *NF1* and *ERBB2*) and Notch signaling (*NCOR1* and *SPEN*) genes. Among these, most driver mutations were specific to either the relapsed tumor or the matched primary tumor, implying that there is typically genomic divergence between primary and relapsed lesions. Moreover, eight out of 10 patients acquired one or two additional driver mutations specific to the relapse sample, suggesting that the growth of the relapsed clone in its new architecture is abetted by further tumor progression, especially in lesions with HRD.

### 2.4. Clonal Architecture BC Progression

To reconstruct the clonal architecture underlying tumor progression from primary lesion to relapse, we applied a non-parametric Bayesian model (PhyloWGS) based on mutant allele fractions of short variants and population frequencies of copy number alterations (CNAs) [25]. In total, we identified an average of 4.6 distinct subclones per sample pair, all sharing a common ancestor. We further stratified these tumors based on the clonal architecture, composition, and genomic feature.

Overall, three major models of clonal architecture from primary lesion to relapse were identified. In patients with model 1, the relapsed clone emerged by acquiring additional mutations from the founder clone of the primary tumor (BC01, BC03, BC06, and BC10; Figure 4A, Appendix A). Typically, most branch mutations in subclones of the primary tumor were eradicated by treatment and were not detected at relapse. Patient BC03 initially harbored a trunk driver mutation in *SF3B1*, composed of two subclones at diagnosis of the primary lesion (Figure 4A). With the entire extinction of these subclones after treatment, we identified alterations occurring in a punctuated evolution [26], leading to relapse, which originated from an additional driver gene, *BRCA2*. Similarly, patient BC01 exhibited punctuated evolution in the primary tumor, where initially a *GATA3* mutation and a *CDK4* copy number gain resulted in the founder clone. Interestingly, in patient BC01, *CDK4* copy number further increased through amplification, which was followed by branching evolution leading to three subclones involved in relapse, with an additional mutation of *ERBB2*. Based on the characteristics of model 1, it will be referred to hereafter as the “*de novo* subclone evolution model (DNSEM)”.

In model 2, compared to the DNSEM, all three cases of relapsed lesions had the HRD phenotype. A dominant relapse-specific clone occurred with a minor subclone of the primary tumor that survived treatment in this model (BC02, BC05, and BC07; Figure 4B, Appendix A), whereas subclones of the primary tumor were eradicated after treatment in DNSEM. A subset of branch mutations persisted after local extinction, with the resultant emergence of both subclonal lineages in tumor maintenance. Therefore, “HRD evolution model (HRDEM)” will be used as the designated name for model 2, henceforth. Patient BC05 carried a germline mutation of *WRN* and shared a *PIK3CA* driver mutation in both lesions, with one subclone evolved from the founder clone to relapse (Figure 4B). The founder clone underwent punctuated evolution and numerous drivers (amplification of *MYC* and *ERBB2*, and *IDH1* mutations) occurred towards branch evolution [26], with new subclones. Moreover, the HRD phenotype was observed in this relapse lesion, probably due to a somatic LOH of *WRN* acquired during tumor progression. Notably, in both models (DNSEM and HRDEM), the HR-related somatic LOHs typically survived in clones of the primary tumor, while the somatic mono-allelic point mutations frequently accumulated in the relapsed lesion. One patient (BC07), a germline *BRCA2* mutation carrier, harbored additional somatic LOH of *BRCA2* in both lesions.

In cases classified as model 3, referred to as the “selective expansion evolution model (SEEM)” hereafter, a minor clone survived to have a selective advantage in the relapsed tumor (BC04, BC08, and BC09; Figure 4C, Appendix A). Specific mutations in a subclone might lead to either endocrine resistance or insufficient treatment post-standard care of BC. Compared to other models, a survived subclone of the primary tumor expanded into the dominant subclone (cancer cell fraction, CCF > 0.45, including its child clones) in the relapsed tumor of SEEM, whereas, in other models, subclones of the primary tumor were survived with lower CCF (<0.15) than other emerged subclones in relapsed tumors or completely eradicated after treatments. Patient BC09 harbored *ERBB2* amplification in a subclone, which exhibited stable growth during relapse with mainly the emergence of a child subclone from punctuated evolution and its branches (Figure 4C). In the case of BC04, we identified an endocrine-resistant mutation of *CTCF*, with the outgrowth of its subclone coupled with the emergence of multiple subclones.

Based on these three models, most relapsed tumors retained some genetic characteristics of the primary breast lesions, but new characteristics became dominant during their evolution. These findings were further corroborated by an analysis of subclonal diversity, derived from the Shannon and Simpson diversity indices [27,28]. Seven out of 10 relapsed tumors exhibited elevated subclonal diversification compared with their primary lesions (Figure 4D); lesions in SEEM showed relatively higher subclonal diversity than those in other models, in both the primary and relapsed states. Notably, patients in SEEM, typically with a subclone of selective advantage, exhibited clonal relatedness between their two lesions, in contrast to cases of HRDEM, which showed an extensive divergence of subclonal diversity in their lesions.

### 2.5. Evolution of the Mutational Processes over Time

Previous evidence suggests that the mutational processes, sculpting the mutational portrait of tumors, may change at different stages of disease evolution [6,12]. Hence, we sought to examine their relative contributions to the transition from the primary tumor to the relapsed state. In this study, we identified three *de novo* single base substitution mutational signatures (*de novo* SBSs) that were enriched in our cohort (Figure 5A).

On average, we found some shifts, over time, in the relative contribution of mutational processes to relapse (Figure 5B). The *de novo* SBS1 (similar to the Catalogue Of Somatic Mutations In Cancer (COSMIC) single base substitution signature 1 (COSMIC SBS1)), enriched for C > T transitions in the NpCpG context, contributed a relatively higher proportion of mutations in primary tumorigenesis. This signature is relatively constant throughout aging and gets swamped by treatment and processes emerging in relapse evolution [29]. The *de novo* SBS2 characterized by C > G transversions and C > T transitions in the TpC context (similar to COSMIC SBS2), attributed to the activity of apolipoprotein B mRNA editing enzyme, catalytic polypeptide-like (APOBEC) DNA cytosine deaminase activity, was more prominent later in the relapsed tumors. Similarly, the *de novo* SBS3 showing high similarity to COSMIC SBS5, with unknown etiology, was predominant at a later time in the relapsed tumor.

We further explored the mutational processes that shaped the three evolution models. In DNSEM, there was a large decrease in the *de novo* SBS1 from the primary tumor to relapse, but an increase in other signatures (Figure 5C). Similarly, patients from HRDEM exhibited a reduction in *de novo* SBS1, accompanied by an indication of *de novo* SBS2 in the relapsed tumors (Figure 5D). In SEEM, the *de novo* SBS2 was dominant in the primary tumor and increased at relapse, suggesting that APOBEC-mediated mutagenesis shaped the whole evolutionary trajectory of both lesions (Figure 5E).

### 2.6. Integration of Actionability and Tumor Evolution in Therapeutic Decisions

To investigate whether additional actionable biomarkers might sensitize primary and relapsed lesions to certain targeted agents and improve treatment choices for future patients, we specifically focused on the driver events of two types: (1) HRD phenotype for PARP inhibitors and/or DNA crosslinking agents [17,30], such as platinum salts, and (2) specific alterations presenting in the genome that are predictive of the response to available Food and Drug Administration (FDA)–approved drugs based on the OncoKB database [31]. We calculated the percentage of primary and relapsed lesions that contained these biomarkers. Nine out of twenty (45%) lesions had at least one actionable event for currently available HRD-related or FDA-approved drugs (Table 2). Among these, apart from a bi-allelic *BRCA2* carrier in both lesions, two additional *BRCA*1/2-wild-type patients were identified as HRD phenotypes in their relapse lesions (Table 2). Moreover, four lesions harbored an *ERBB2* amplification, all of which were clinically identified as HER2− (Table 2). These lesions might benefit from the approved anti-HER2 therapies for BC. Two lesions might be treated by the recently FDA-approved BC drug to target *PIK3CA*. Additionally, two patients had one alteration predictive of a response to a drug registered for other tumor types (not BC). Notably, actionable biomarkers tended to occur in relapsed lesions rather than primary lesions.

Importantly, the three evolution models strongly support the use of genome sequencing of BC lesions to consolidate therapeutic decisions. For patients from DNSEM, most subclones progressed via punctuated evolution. Once actionable alterations appeared, they were present in nearly all the tumor cells at presentation and after relapse, suggesting that patients might benefit from the targeted drugs at both early and later stages. Interestingly, for patient BC03, an actionable *BRCA2* mutation might also indicate PARP inhibitors as a potential treatment choice in the relapsed lesion, as the cancer cell fraction of this mutation was 94.3% (Table 2). Although the relapse sample of BC03 had a higher HRD score compared to the matched primary one, the benefit from PARP inhibitors was not clear due to its HRD score <42. Since the HR defect was uniformly present in clones of patients from HRDEM, the relapsed lesions could be considered as potentially treatable with therapies that induce DNA double-strand breaks, such as PARP inhibitors and/or platinum-based agents [17,30]. However, patient BC07 with an HRD score <42 might not have the predictive response to such therapies. Moreover, patient BC05 had several actionable alterations in both the founding clone and its subclone, suggesting combination therapy regimens in the adjuvant setting. Subclones of SEEM frequently evolved from branch nodes that made them unsuitable to be treated by these established therapies. For example, in patient BC09, multiple subclones were derived from an actionable *ERBB2* amplification in both lesions with only 55.4 and 68.2% of cancer cells.

## 3. Discussion

Understanding how primary tumor cells evolve into relapse is critical to understanding tumor progression and identifying therapeutic opportunities in BC patients. Our longitudinal genomic characterizations uncovered features such as the accumulation of deficiency in the HR pathway, with increased CIN and divergence of somatic drivers. These observations could potentially illuminate the individualistic paths of relapse after treatment. In this study, we reconstructed the clonal architectures of 10 patients with relapse in order to understand treatment failures and major evolutionary patterns. Our findings highlighted the changing mutagenic stresses over time and clinically relevant molecular features that can inform therapeutic choices across three evolutionary trajectories.

In this study, we observed that four out of 10 patients had a notably increased HRD score in their relapsed tumors, compared to matched primary ones. Among these patients, there were two relapsed tumors with an HRD score ≥42 that could be treated and might have a predictive response to PARP inhibitors and platinum-based agents [17,30,32]. In the other two cases, such treatment might also be the potential therapeutic avenue, particularly in the case of BC03 with the somatic *BRCA2* mutation. Patient BC07 with HRD phenotype harboring the bi-allelic *BRCA2* alteration could also be considered for these treatments. However, the efficacy of PARP inhibitors and platinum compounds was unclear in these two patients with an HRD score <42.

Our results reveal that not all relapsed lesions have the same somatic drivers as their primary tumors. Although the genome of a relapsed clone might be similar to that of the primary lesion at initial diagnosis, by the time it had been expanded to be detectable by sequencing, extreme genomic divergence had occurred. Instead, a higher number of somatic drivers was found that were specific to relapsed tumors. There are two possible explanations for the enrichment of drivers in relapsed lesions compared to the primary cancer cohort: (1) relapsed lesions with a substantially increased rate of mutation accumulation are more likely to acquire drivers; or (2) the HRD phenotype contributes to the acquisition of *de novo* driver mutations in relapsed clones. Most tumors show rather distinct combinations of driver mutations, with *GATA3* and PI3K-pathway mutations characterizing lesions. Furthermore, the tumor cells surviving via endocrine-resistant mutations do continue to evolve after treatment, probably by expansion of the dominant clone.

Furthermore, our analyses found that the progression of primary lesions to relapse proceeds in accordance with three major evolution models. Previous reports provided evidence that subclonal diversification of a tumor is directly linked to the aggressiveness of the disease [33], while the difference of subclonal diversification between different lesions indicates their clonal relatedness [34]. Patients classified as DNSEM with subclones from punctuated evolution in both lesions might have remarkably good outcomes. We observed that patients whose evolution followed the DNSEM pattern did not always require clonal selection and exhibited low clonal relatedness (a high difference in subclonal diversification) between primary and relapsed lesions. For HRDEM, HR-related somatic LOHs might be involved in primary tumor maintenance, while the accumulation of additional HR-related alterations might induce a switch to branch evolution and the acquisition of somatic drivers leading to relapse. Importantly, these genomic events produce the HRD phenotype in the relapsed state. Moreover, these patients showed relatively elevated subclonal diversification from primary tumors to relapsed tumors compared with other models. SEEM created subclonal diversity via branch evolution, and a subclone typically survived post-therapy at the original site. It is possible that high subclonal diversity increases the probability of a clone’s capability to evade treatment in the primary lesion. The surviving tumor subclone expands to form its child clones, and subsequent evolutionary branches compete and collaborate in the relapsed lesion. This might explain the low difference in subclonal diversification between primary and relapsed tumors in this model. In conclusion, our observations regarding these evolution models uncovered distinct genomic characterizations during progression.

We further observed a shift from enriched age-related mutational processes in primary tumors towards more APOBEC-driven mutagenesis or underlying endogenous mutagen exposures in relapsed lesions. This observation was quite varied in the three evolution models. Notably, APOBEC-driven mutagenesis might shape the whole evolutional trajectory in SEEM.

Our results showed that several potential targeted agents might provide clinical benefit for BC patients with worse outcomes. HER2− patients with *ERBB2* alterations might benefit from approved anti-HER2 therapies. As existing actionable biomarkers could be confounded by resistant mutations/clones, and their respective subclones or/and evolutionary branches, new therapeutic regimens are necessary. Furthermore, targeting alterations in both parent and child clones would possibly be expensive and lead to overtreatment. Our study provides a framework for the integration of clonal architectures and CCF into therapeutic decisions to define patients with a greater likelihood of being cured. This framework led to both a retrospective explanation of previous treatment failure in the primary tumor and a prospective prediction of therapies that could prevent relapse. Being able to identify the actionable biomarkers in the founder clone could facilitate treatment strategies for early intervention towards the arrest of progression and avoidance of relapse [35], especially in DNSEM. Cases of HRDEM with HRD scores ≥42 would probably respond to PARP inhibitors and platinum compounds [17,30,32], and future clinical studies of relapse could include this biomarker as a covariate in the data analysis. This phenotype that frequently has an increased number of drivers could also be effectively exploited to identify targeted agents. In SEEM, targeting somatic alterations found in this clone would not eventually be sufficient to cure the relapse. Moreover, the expansion of a clone carrying resistant alterations likely reflects the clonal replacement or selective advantage operating by therapeutic interventions during the branch evolution of relapsed BC. Taken together, clinicians could consider employing this framework for future improvement of patient management, such as intensification/de-intensification of treatment for curable cancers.

There are some limitations that exist in this study. Although we found no direct correlation between differences in the purity and differences in the HRD score and CIN level, it should be noted for future studies. Furthermore, we observed a positive correlation between CIN and HRD. However, it was not the only mechanism to cause CIN, which could also be influenced by whole-genome doubling. Furthermore, the COSMIC substitution signature 3, which related to HRD, was not identified. However, in previous studies, this signature did not provide a precise assessment of HRD [8,36]. Although some BC samples had low numbers of mutations, we could still extract and uncover the change of predominant mutational signatures (age and APOBEC signatures) from primary tumors to their relapse. We believe that collecting more IBTR samples in future studies would provide further valuable views for understanding the changes in mutational stresses over time and help to study the relationship between time to relapse and subclonal diversification.

## 4. Materials and Methods

### 4.1. Study Population and Specimens

Formalin-fixed and paraffin-embedded (FFPE) samples from tumor and matched adjacent normal tissues in a cohort of 10 patients with BC were collected. All the relapse samples were collected from these same patients. Subjects recruited for this study included a subset of BC patients with Taiwanese ethnicity who were diagnosed and received surgical resection at Lotung Poh-Ai Hospital (Yilan County, Taiwan) and Cathay General Hospital (Taipei, Taiwan) in Taiwan. The study protocols were reviewed and approved by the institutional review boards of these hospitals, and informed consent was obtained from all patients for conducting WES and corresponding analyses. BC subtypes were identified using immunohistochemistry and fluorescence in situ hybridization. The clinicopathological characterization of the patients and drug information were collected from inpatient medical records (Appendix A).

### 4.2. Exome Capture, Library Construction, and Sequencing

For the generation of standard exome capture libraries, we used the Agilent SureSelect XT Reagent kit protocol (Agilent, Santa Clara, CA, USA) for an Illumina Hiseq paired-end sequencing library (catalog#G9611A; Illumina, San Diego, CA, USA). In all cases, the SureSelect XT Human All Exon Version 6 (60 Mb; Agilent, Santa Clara, CA, USA) probe set was used. We used 1000 ng genomic DNA to construct each library. Each adapter-ligated sample was purified using Agencourt AMPure XP beads (Beckman Coulter Life Sciences, Indianapolis, IN, USA) and analyzed on a Bioanalyzer DNA1000 chip (Agilent, Santa Clara, CA, USA). A total of 750 ng of the sample was prepared for hybridization with the capture baits, and the sample was hybridized for 90 min at 65 °C, captured with Dynabeads MyOne Streptavidin T1 beads (Thermo Fisher Scientific, Waltham, MA, USA), and purified using Agencourt AMPure XP beads (Beckman Coulter Life Sciences, Indianapolis, IN, USA). We used the Agilent protocol to add index tags by post-hybridization amplification. Finally, all samples were sequenced on an Illumina Hiseq4000 instrument (Illumina, San Diego, CA, USA) using the 150PE protocol. The average sequencing depth of WES was more than 200 in both tumor and matched normal tissues.

### 4.3. Sequencing Data Processing and Quality Assessment

Illumina library adapter sequences from paired-end FASTQ files were trimmed using Trimmomatic (v.0.33) to enhance the quality of downstream alignment [37]. The processed Binary Alignment Map (BAM) files were aligned to the UCSC human reference genome (hg19) using Burrows-Wheeler Aligner-MEM (BWA-MEM) (v.0.7.15) [38]. The Picard module (v.2.6.0) (https://broadinstitute.github.io/picard/; accessed on 8 August 2016) was utilized to sort BAM files containing the sequence alignment data in binary format. Duplicate reads were marked for exclusion in subsequent analysis using the MarkDuplicates tool in Picard. Base quality score recalibration was conducted to assign an accurate confidence score to each base using known variants in dbSNP138, the Mills and 1000G gold standard INDELs [39,40]. To improve downstream variant detection, we added 100 bp-interval padding to confirm all reads within and outside the targeted region.

### 4.4. Variant Calling and Post-Processing Strategies

For the variant calling pipeline, the following five methods were applied. Mutect2 (v.4.1.0) was used to conduct both SNV and INDEL calling in targeted exons, utilizing gnomAD as the reference to assess known germline events [41]. Following the Genome Analysis Toolkit (GATK) Best Practices workflow (v.4.1.0) [39], we filtered somatic mutations by: (1) removing false-positives, systemic bias, and germline mutations using a previously described set of 124 non-tumor normal samples in the Taiwan population (Panel-of-Normals (PoN) filter); (2) removing 8-oxoguanine DNA lesions and FFPE artifacts caused during sequence library preparation; and (3) removing bias from cross-sample contamination by DNA. SNVs and INDELs were also identified by Strelka2 (v.2.9.10) and filtered based on its empirical variant scoring (EVS) model (removing “LowEVS” calls) [42]. We further used the following popular callers to identify somatic SNVs: (1) MuSE (v.1.0rc), to obtain the stringent “pass” calls by tier-based cutoffs (removing tier 1 to 5 calls) from a unique sample-specific error model [43]; (2) SomaticSniper (v.1.0.5.0), to make alteration calls by implementing the genotype likelihood model of Samtools MAQ [44]; and (3) Varscan2 (v.2.3.9), to detect somatic calls based on a heuristic and statistical algorithm [45]. Default parameters were used for these callers. Except for Mutect2, similar to the previous study [46], we applied additional post-hoc filters to remove false-positive mutations. The first of these, “fpfilter”, was employed by major sequencing centers, including TCGA, to remove calls from some classes of systematic sequence error (https://github.com/ckandoth/variant-filter; accessed on 31 May 2017). Another, “filter_ffpe”, employed by Memorial Sloan Kettering Cancer Center (New York, USA), was utilized to filter mutations stemming from the formaldehyde deamination of cytosines (https://github.com/mskcc/ngs-filters; accessed on 10 April 2019). Finally, to obtain the most robust calls, only SNVs reported by at least two callers and INDELs identified by both Mutect2 and Strelka2 were used. In addition, we also filtered mutations for further analysis by: (1) removing SNVs that met the non-pass Mutect2 criteria; (2) removing mutations within non-targeted regions; (3) removing variants found in the Taiwan Biobank database with a population allele frequency > 0.0001 [47]; (4) removing mutations with population allele counts ≤ 10 across at least one Exome Aggregation Consortium non-TCGA subpopulation (v.0.3.1) [48]; and (5) filtering potential FFPE artifact INDELs with bias in read pair orientation as previously described [49]. The DeTiN (v. 1.8.9) algorithm was employed to rescue the rejected calls due to potential tumor-in-normal contamination [50]. Somatic mutations were annotated, and a mutation annotation format (MAF) file was produced using Variant Effect Predictor (VEP v.89) [51] and a vcf2maf.pl script (https://github.com/mskcc/vcf2maf; accessed on 2 May 2019).

### 4.5. Subclonal Copy Number Assessment

Subclonal copy numbers were called using the FACETS algorithm (v.0.5.2) for WES data [52]. Reference and variant allele read counts were extracted from the tumor and matched normal BAM files at germline polymorphic sites, which are catalogued in the dbSNP and 1000 genomes databases with base quality > 20 and mapping quality > 15, and only sufficiently covered regions with > 25 read counts were considered. Heterozygous single-nucleotide polymorphisms in normal samples were used, and allele-specific copy number profiles for matched tumor samples were analyzed with default settings. Tumor purity and ploidy were utilized to obtain accurate allele-specific copy number profiles and facilitate the identification of subclonal copy number events. Arm-level and gene-level CNAs were distinguished by using GISTIC 2 (v.2.0.23) [53] with the default parameters, and the clonal assessment was based on the associated cellular fraction estimates from FACETS.

### 4.6. Large-Scale Genomic Events Analysis

Three large-scale genomic events were identified across tumor samples and are described below in detail.

#### 4.6.1. Homologous Recombination Deficiency

A genome aberration-based scoring system (HRD score) derived from the unweighted sum of the HRD-LOH score [54], the tAI [55], and the LST score [56], was employed to assess the underlying tumor HRD [17]. The HRD-LOH score is defined as the number of LOH events > 15 Mb without covering the whole chromosome. The tAI score is the number of regions with allelic imbalance, which were extended towards the telomeric ends of a chromosome without crossing the centromere. The LST score is the total number of breakpoints between regions of at least 10 Mb, with a distance between them shorter than 3 Mb. Based on allele-specific copy numbers for each region, the extent of HRD was quantified by using scarHRD (v.0.1.0) [57]. A predefined HRD threshold of ≥42 was employed to distinguish the HRD phenotype from nondeficient tumors [17].

#### 4.6.2. Chromosomal Instability

CIN is a broad concept that encompasses a wide range of chromosome-level abnormalities. The level of CIN is defined as the percentage of the genome affected by CNAs [10].

#### 4.6.3. Whole-Genome Doubling

Patients were considered to have undergone WGD if greater than 50% of their autosomal genome had a major copy number derived from FACETS greater than or equal to two.

### 4.7. Assessment of Bi-Allelic Alterations in HR-Related Genes

A published list of 102 core and related HR genes was employed for assessment as previously described [19]. LOF mutations were defined as those with a clearly functional impact on a gene, including frameshift, nonsense, start/stop codon changes, and splice site mutations, whereas missense mutations were considered as variants of uncertain/unknown significance (VUS). Germline short-variant discovery was conducted using HaplotypeCaller with the gvcf mode from GATK v.3.7 [39]. The bi-allelic alterations in HR genes refers to cases where both alleles of an HR-related gene were lost, including (1) a germline LOF mutation with somatic LOH of the wild-type allele, (2) a germline LOF mutation with a somatic LOF mutation in the same gene, (3) a somatic LOF mutation with somatic LOH of the wild-type allele, (4) two somatic LOF mutations in the same gene, and (5) a somatic deep deletion. Similarly, we considered (1) a germline LOF mutation with a somatic VUS, or (2) a somatic LOF mutation with a somatic VUS, or (3) a somatic VUS with somatic LOH of the wild-type allele to be a bi-allelic VUS event. Bi-allelic VUS in HRD cases without other bi-allelic LOF alterations were considered putative LOF events.

### 4.8. Potential Driver and Actionable Gene Identification in Primary and Relapsed Lesions

To investigate the potential driver landscape for primary and relapsed lesions, two categories of alterations were retained: missense and LOF mutations. We manually curated the union of significantly mutated genes reported in five previous large cohort BC studies [6,8,21,22,23]. Thus, a total of 90 genes were considered as potential driver genes for further analysis (Appendix A). In addition, the known actionable genes were summarized based on the annotation of levels 1 and 2 of evidence in the OncoKB database [31], and the endocrine-resistant genes were selected based on a previous study [24].

### 4.9. Reconstruction of Cancer Clonal Architecture

The evolution of tumor genetics was reconstructed using PhyloWGS (v.1.0-rc2) [25]. The high-confidence SNV calls and the predicted CNA segments with tumor cellularities were used to build up the PhyloWGS inputs. Using its default settings, PhyloWGS constructed possible architectures for each patient. The best consensus tree was determined by the largest log likelihood value, and SNVs and CNAs were simultaneously assigned to the associated subclones in the best predicted architecture. The visualization of clonal architectures was attained by using the fishplot (v.0.5) R package [58].

### 4.10. Trunk and Branch Alteration Classification and Clonal Architecture Labeling

We classified both SNVs and CNAs as occurring in the trunk or branch of each tree. Trunk alterations encompassed those that appeared between the root node (normal clone) and its only founder clone, while branch alterations were defined as those that appeared in the child nodes of the founder clone. The definition of trunk and branch alterations was employed by using the consensus trees inferred from PhyloWGS. To label the clonal architectures for each patient, we selected six categories of genomic alterations/events that could be classified as trunk or branch, including HR-related bi-allelic alterations, HRD phenotype, driver gene mutations, endocrine-resistant gene mutations, high-confidence CNAs, and actionable alterations.

### 4.11. Subclonal Diversity Assessment

To quantify the subclonal diversity of each sample, we employed the Shannon diversity index (H) and the Gini–Simpson index based on the richness (the number of subclones present) and abundance (the proportion of all clones represented by a single subclone) of subclones [27,28]. The Shannon diversity index originated from information theory to summarize the diversity of a population and is defined as $H\;=\;-\overset{n}{\underset{i=1}{\sum }}p_{i}\ln p_{i}$, where $p_{i}$ is the abundance of clones i, and n is the number of subclones. The Gini–Simpson index is defined as the probability that two entities taken at random from the dataset of interest represent the distinct types by the equation $D\;=\;1\;-\;\overset{n}{\underset{i=1}{\sum }}p_{i}^{2}$, where $p_{i}$ is the abundance of clone i and n is the number of subclones. Clonal abundance was calculated using the CCF of each node derived from PhyloWGS, after subtracting the total CCF of its children.

### 4.12. Mutational Signature Analysis

The final portrait of mutations was determined by the duration of exposure to each mutational process in patients. Mutational signatures were deciphered from the substitution context defined by 96 mutation types, composed of flanking bases as triplet motifs. The contributions of each mutational signature were quantified to deduce its association with mutagenic processes, such as ultraviolet light exposure, carcinogens, and aging. To characterize the new mutational signatures originating from the accumulation of historic mutagenic activity in primary and relapsed lesions, we utilized non-negative matrix factorization (NMF) to extract *de novo* mutational signatures present in these tumors by utilizing the R package NMF (v.0.21.0) [59]. Motif matrices were extracted using MutationalPatterns (v.1.2.1) [60]. The inferred mutational signatures were then compared to the curated catalog of 49 mutational signatures (https://cancer.sanger.ac.uk/cosmic/signatures; accessed on 15 May 2019) referenced in COSMIC using cosine similarity [29,61]. Most signatures with cosine similarity > 0.70 corresponded to one or a mixture of known signatures. To identify genes related to the novel signature, we then utilized linear regression to calculate the correlation between signature exposures and SNVs. The Benjamini–Hochberg method was used to control the false discovery rate due to multiple testing [62].

### 4.13. Statistical Analysis

HRD and CIN were analyzed using a one-tailed paired Mann–Whitney U test to determine if relapsed lesions had higher HRD and CIN than their primary tumors, and the association between HRD and CIN was evaluated by Pearson correlation. All statistical analyses were performed using R version 3.5.1.

## 5. Conclusions

Overall, we have identified the differences in genetic features between primary tumors and their corresponding relapsed lesions, including HRD, CIN, and somatic mutation drivers. Our analyses highlighted major clonal architectures that shape our perspective on mutagenic processes and subclonal diversification during relapse progression. Although our study only involved 10 patients, this framework provides substantial insight into the biology of relapsed BC and integrated potential therapeutic opportunities with clonal architectures towards the improvement of future treatment decisions. We expect that continuing to profile relapsed tumors using comprehensive sequencing approaches will identify other common evolutionary pathways and inform new therapies.

## Figures and Tables

**Figure 1 cancers-13-01821-f001:**
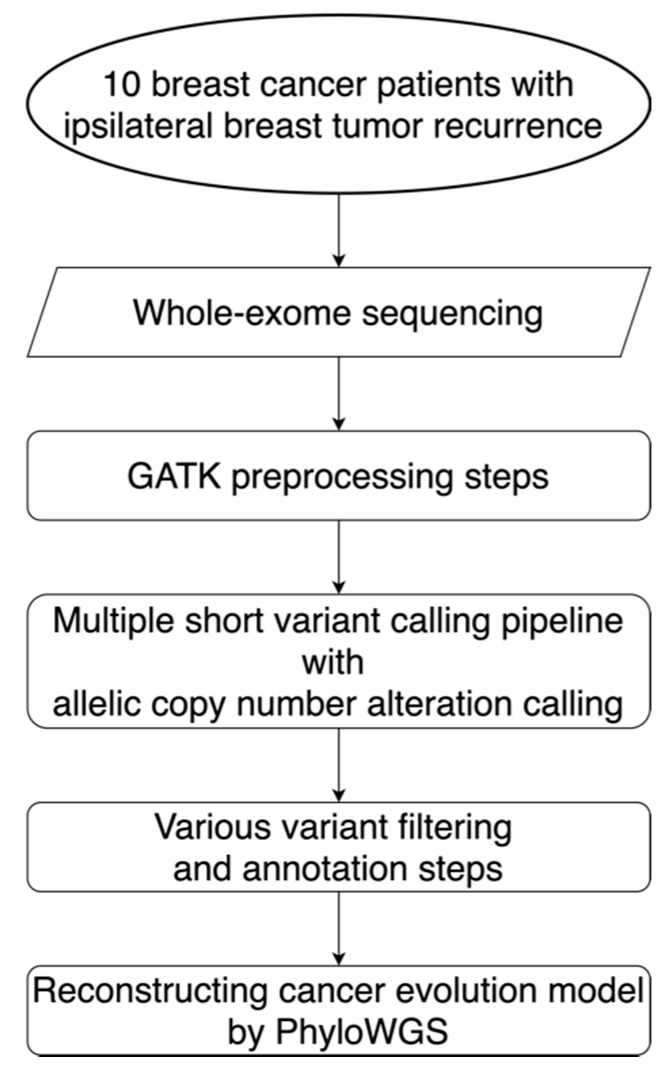
Workflow for the alteration detection and evolution model reconstruction. This workflow diagram describes the design for the sequencing, data preprocessing steps, the alteration calling pipeline, filtering steps, and the reconstruction of evolution models for 10 localized breast cancer patients with ipsilateral breast tumor relapse. GATK: Genome Analysis Toolkit; PhyloWGS: the subclonal reconstruction algorithm for cancer evolution.

**Figure 2 cancers-13-01821-f002:**
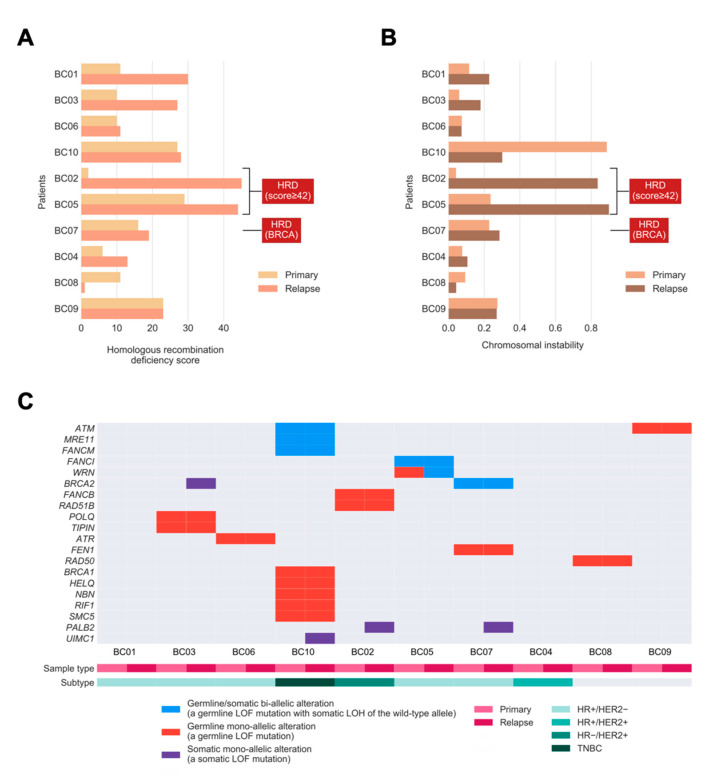
Homologous recombination deficiency (HRD), chromosomal instability (CIN), and bi-allelic/mono-allelic alterations affecting HR-related genes in primary and relapsed cancers. (**A**) Bar chart showing the extent of HRD in relapsed tumors versus primary breast tumors. (**B**) Bar chart showing the level of CIN in relapsed tumors versus primary tumors. The labels indicate tumors with the HRD phenotype, defined as high HRD score (≥42) or/and bi-allelic alterations in *BRCA1/2*. (**C**) Oncoprint showing the incidences of bi-allelic and mono-allelic alterations of HR-related genes in primary and relapsed tumors, stratified according to germline or/and somatic origin. The bi-allelic and mono-allelic alterations are also annotated by colors. LOF: loss-of-function; LOH: loss of heterozygosity; TNBC: HR−/HER2−.

**Figure 3 cancers-13-01821-f003:**
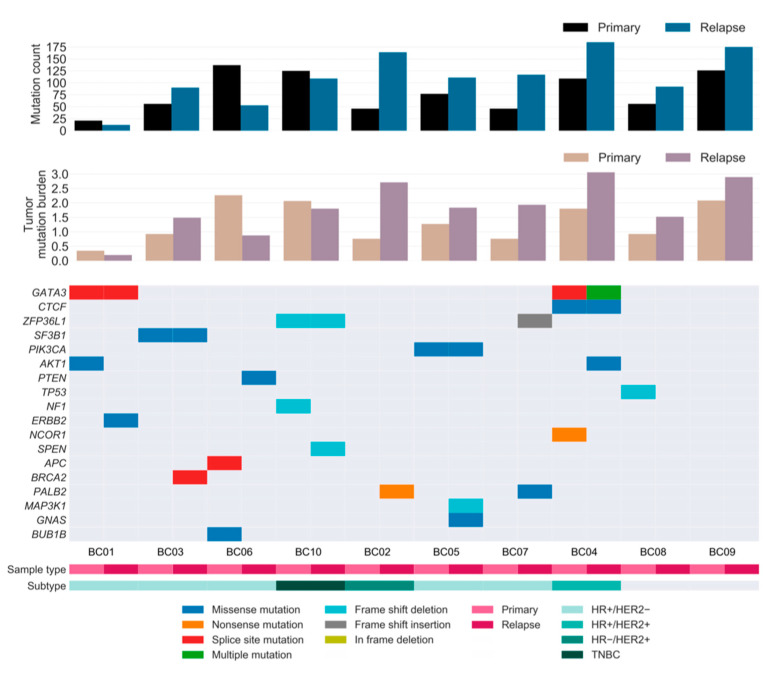
Repertoire of somatic alterations of primary and relapsed cancer samples. Rows represent curated driver genes of breast cancer, and columns represent individual tumors stratified by sample type. Clinical characteristics and the effects of the somatic mutations are color-coded according to the legend. The paired bar graphs (top and middle) depict the mutation count (*y*-axis) and tumor mutation burden (mutations/number of covered bases; y-axis) for primary and relapsed tumors (*x*-axis), respectively. TNBC: HR−/HER2−.

**Figure 4 cancers-13-01821-f004:**
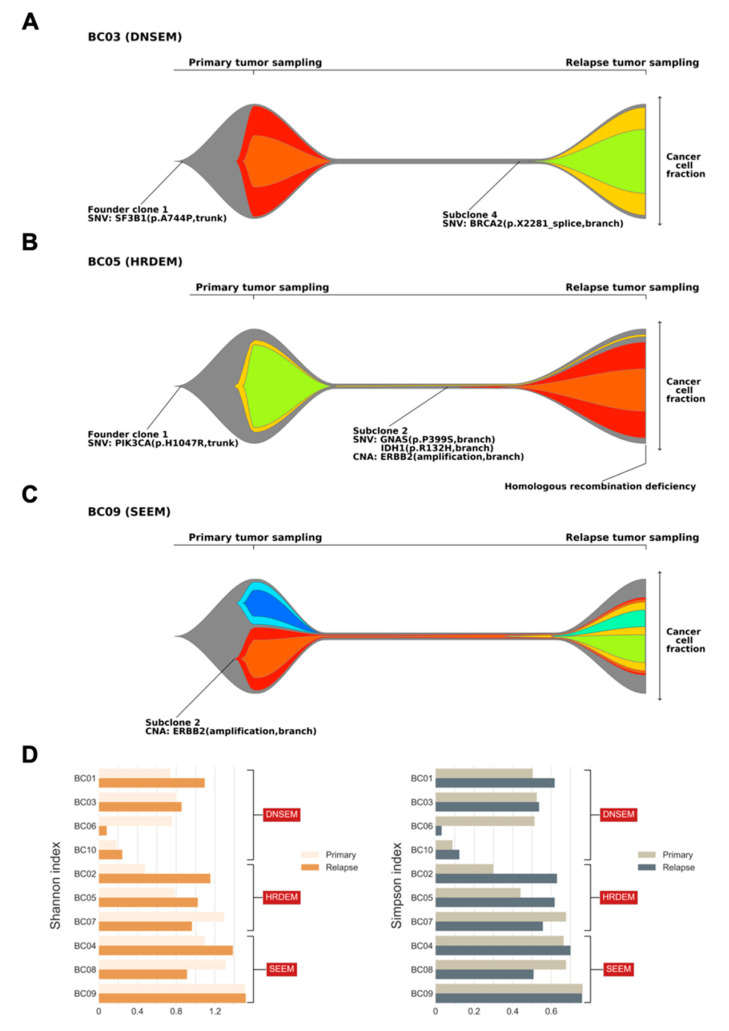
Clonal architecture of primary and relapsed tumors, and indices of subclonal diversity. Fishplots indicate the clonal composition and the potential clonal selection during progression of primary tumors to relapse from three patients ((**A**) BC03; (**B**) BC05; and (**C**) BC09). Each plot was derived from somatic single nucleotide variations (SNVs), insertions/deletions, and copy number alterations (CNAs) using PhyloWGS. The founder clones are shown in gray, while subclones are shown in other colors. Key alterations in the founder clone and subclones are highlighted, with mutations in driver genes identified at both the clonal and subclonal level. Tumors with the homologous recombination deficiency phenotype are also annotated. The cancer cell fraction of each clone could be inferred at the time of primary and relapse sampling. Three major models of clonal architecture from primary lesion to relapse were identified (DNSEM: *de novo* subclone evolution model; HRDEM: HRD evolution model; and SEEM: selective expansion evolution model). (**D**) Bar charts showing the subclonal diversity in relapsed tumors versus primary breast tumors, derived from two indices (left: Shannon index; and right: Simpson index).

**Figure 5 cancers-13-01821-f005:**
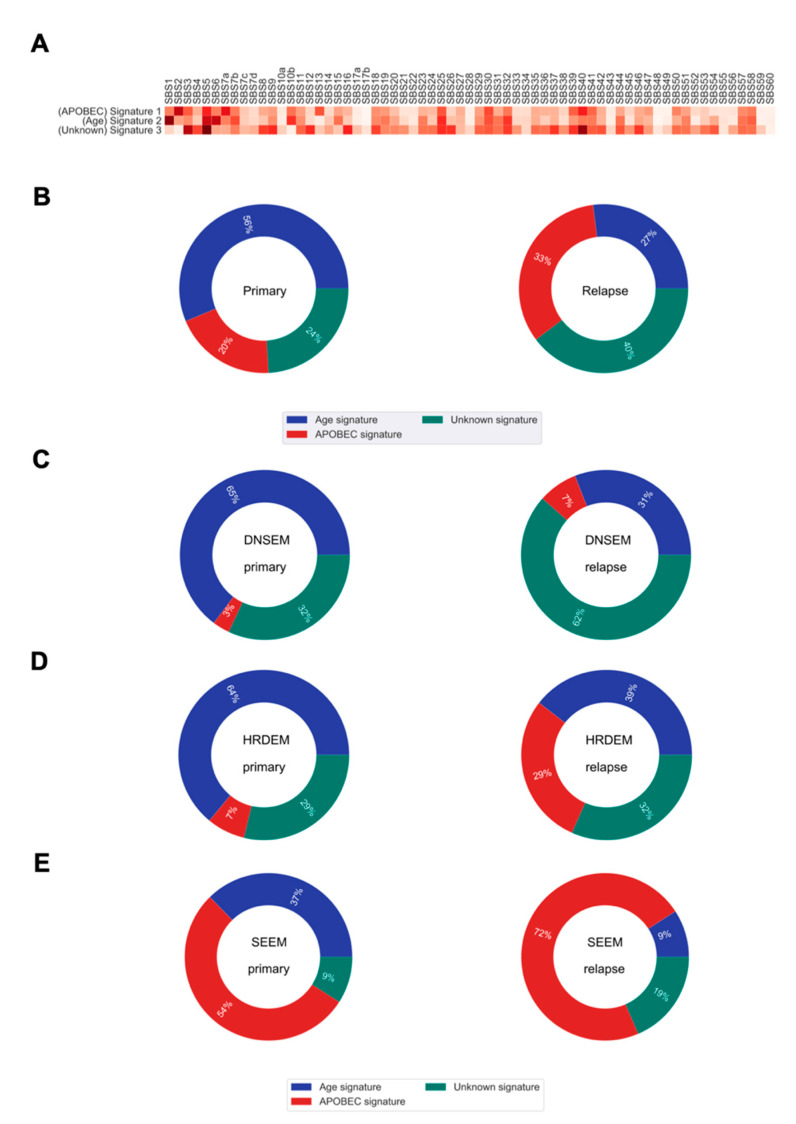
The mutational signatures and their contributions to somatic mutations in primary and relapsed tumors. (**A**) Heatmap of the cosine similarity results for the three *de novo* single base substitution signatures of our cohort (*y*-axis), coded by color. The cosine similarity (range 0–1) represents the extent of similarity to a particular signature of the Catalogue Of Somatic Mutations In Cancer (COSMIC) (*x*-axis). Among the 65 COSMIC mutational signatures, the age and apolipoprotein B mRNA editing enzyme, catalytic polypeptide-like (APOBEC) signatures were the most similar mutational signatures detected in both primary and relapsed tumors (dark red), while one signature was recorded as unknown etiology in the COSMIC database. (**B**) Pie charts show the percentage of mutations derived from each signature, for both the primary and relapse cancer cohorts. (**C**–**E**) Pie charts show the percentage of mutations derived from each signature for each of the three evolution models from primary tumor to relapse. DNSEM: *de novo* subclone evolution model; HRDEM: HRD evolution model; and SEEM: selective expansion evolution model.

**Table 1 cancers-13-01821-t001:** Clinicopathologic characteristics of 10 patients.

Case ID	Age at Diagnosis (Years)	Laterality	Grade	Tumor Size (Invasive, mm)	Lymph Node Status	ER	PR	HER2	Subtype	Relapse after Primary Diagnosis (Months)
BC01	48	Right	1	15	−	+	+	−	Luminal A-like	
BC02	70	Left		<1	−	−	−	+	HER2-enriched	6
BC03	39	Left	2	40	−	+	+	−	Luminal A-like	45
BC04	47	Left	2		−	+	+	+	Luminal B-like	61
BC05	70	Left	2	18	−	+	−	−	Luminal A-like	59
BC06	29	Right	1	47	−	+	+	−	Luminal A-like	
BC07	36	Left	2	24	−	+	+	−	Luminal A-like	
BC08	47	Right	2	13	−	−	−			24
BC09	72	Right	2	25	−	+	−			23
BC10	27	Right	2	32	−	−	−	−	Triple negative	107

Blank cells indicate missing data. Luminal A-like: hormone receptor-positive (HR+)/human epidermal growth factor receptor-negative (HER2−); Luminal B-like: HR+/HER2+; HER2-enriched: HR−/HER2+; triple-negative: HR−/HER2−.

**Table 2 cancers-13-01821-t002:** Therapeutic actionability.

Sample ID	Actionable Biomarker	Actionable Alteration	Clone	Cancer Cell Fraction	Classification
BC01 (relapse)	OncoKB	*CDK4* amplification	Clone 1	1	Trunk
BC01 (relapse)	OncoKB	*ERBB2* mutation	Clone 5	0.522	Branch
BC02 (relapse)	HRD	(HRD score ≥ 42)	N/A	N/A	N/A
BC03 (relapse)	OncoKB	*BRCA2* mutation	Clone 4	0.943	Branch
BC05 (primary)	OncoKB	*PIK3CA* mutation	Clone 1	1	Trunk
BC05 (relapse)	OncoKB	*PIK3CA* mutation	Clone 1	1	Trunk
BC05 (relapse)	OncoKB	*IDH1* mutation	Clone 2	0.838	Branch
BC05 (relapse)	OncoKB	*ERBB2* amplification	Clone 2	0.838	Branch
BC05 (relapse)	HRD	(HRD score ≥ 42)	N/A	N/A	N/A
BC07 (primary)	HRD	Bi-allelic *BRCA2*	N/A	N/A	N/A
BC07 (relapse)	HRD	Bi-allelic *BRCA2*	N/A	N/A	N/A
BC09 (primary)	OncoKB	*ERBB2* amplification	Clone 2	0.554	Branch
BC09 (relapse)	OncoKB	*ERBB2* amplification	Clone 2	0.682	Branch

N/A: Not applicable; HRD: homologous recombination deficiency; OncoKB: a precision oncology knowledge base.

## Data Availability

The raw sequencing data (FASTQ files) in this study are available on request from the corresponding author. The FASTQ files are not publicly available due to privacy reasons.

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
