# Peer review of "Evolutionary Trajectories and Genomic Divergence in Localized Breast Cancers after Ipsilateral Breast Tumor Recurrence"

_cancers, 2021, doi:10.3390/cancers13081821_

Round 1

Reviewer 1 Report

The manuscript of Wu et al. tried to describe how various IBTR evolve and suggest potential treatment applications. I have minor comments regarding the presentation of the study. 

1. The study would really benefit from adding a workflow figure after the introduction. It should include that 10 patients with IBTR were analysed, then exome sequencing, then the various filtering and quality control step and the model building at the end. 

2. It was not exactly clear for me that the model building step was three different methods fo model building or it was three different models of metastasis clonal selection. After checking it is the latter. Please add a short figurative description to the current Figure 3. 

3. Figure 1 A miss annotated the high HRD patients. BC03 and BC02 had high HRD score but on the figure BC02 and BC05 has. 

4. Figure 4 A What is the APOBEC signature? Is it a gene? 

5. Figure 5 is misleading. From 10 patients using such a pie chart makes the results really confusing. I suggest to remove it because the text explains the results and therapeutic indications.

6. I suggest to tone down a bit the discussion. The authors analysed only 10 patients and the results are really valuable but observational. 

Reviewer 2 Report

Interesting study looking to explore clonal divergence between a primary tumour and matched ipsilateral recurrence by exome sequencing. The authors explored 10 cases.

Some comments below, in chronological order, some are minor points, others need careful consideration and re-evaluation.

  1. Line 96: “majority” should read ‘half’
  2. Table 1: time to relapse would be important to get for cases where it is missing.
  3. Section 2.2 – has lots of confusing elements:
    1. The differences in HRD scores is quite striking for some cases, but the authors should calculate the tumour cellularity of the samples sequenced. They should be able to do this using the WES data. This is important . if the cellularity of the primary and matched recurrence specimens are different then this may explain variation in for eg HRD and CIN scores ( plus other things such as mutations identified, distribution of mutation signatures). This is pertinent for cases 1, 3, 6,10, 2, 4 where there are differences in HRD score between matched tumours.
    2. Whilst some of these cases do show an increase in score in the recurrence, the scores are still often below the 42 threshold and so the samples remain HR proficient, by this cut off.
    3. although the CIN and HRD have a high correlation (line 122) there are quite striking differences between these metrics for some cases (eg 1, 3, 10, 5, 4) so I don’t agree with statement on lines 124-5, or 128.
    4. Line 127 is wrong – case 5 does not have a high HRD score.
    5. Fig 1C is confusing, I think in part as the legend must be wrong. Should one of the colours blue, red or purple reflect somatic LOH? Or does the fig not show biallelic LOF? When you say (line141) that 3/10 tumours had at least 1 shared biallelic LOF in an HR gene – does this refer to cases 10, 5 and 7? Are these then referring to germline biallelic alterations? How does that work? Mostly one sees a germline pathogenic plus a somatic second hit. Case 7 for instance – does it harbour 2 germline pathogenic variants in BRCA2? are these on the same or different alleles? but this case doesn’t show HRD or CIN. Case 5 – how can you explain in the primary tumour it has a germline monoallelic pathogenic, but in the recurrence it has a biallelic germline pathogenic? The germline can not change. Caes 10 – I find it hard to believe this tumour carries so many germline biallelic & monoallelic pathogenic mutations. Can you describe these variants and verify they are pathogenic through variant databases such as clinvar?
    6. Line 142 – what do you mean by a 20% higher rate versus TCGA? Please explain this more.
    7. Line 145 – says case 5 has an increase in HRD in the recurrence. Id prefer you say it had an increase in CIN. I suspect HRD isn’t the only mechanism to cause CIN
    8. Put this into context – 2 cases had an increase in HRD (going from proficient to deficient based on the scores). 1 additional case had an increase in CIN. The so called BRCA2 deficient case was based on biallelic germline variants, and didn’t show HRD.
  4. Section 2.3.
    1. Line 162 – are ZFP3L1 and SF3B1 well known breast cancer genes?
    2. Line 164 – describes a GO analysis but shows no data. How many mutations were used to do this analysis. Do you have power for this to be robust. I would remove this.
    3. line 168 - The mutations mentioned were ‘enriched’ in the cohort – compared to what? How is AKT/PTEN enriched when only 2 and 1 case had a mutation, respectively.
    4. Fig 2 – would be very useful to show here the absolute # of mutations identified in a bar plot above the mutation burden plot. Presenting these numbers is also v important for other aspects of the study such as signatures.
  5. Section 2.4
    1. I kind of like this sort of analysis but I don’t know how much power you have to perform it when the #s of mutations used isn’t described and you only have 10 cases – is this sufficient?. Have these 3 models been previously described for validation of your work? How does the population frequency of CNAs help in the calculations.
    2. Cases 2, 5 and 7 were in the HRDEM model, yet 5 and 7 don’t show HRD and case 3 does but isn’t featured in this model.
  6. Section 2.5 – signatures
    1. Interestingly HRD substitution signature is not identified
    2. Again it would be great to know the numbers of mutations used to call these signatures in each case, so we can determine the robustness of the classification and the biological interpretation. Some breast cancers have quite low numbers of mutations (as noted in your TMB plot in Fig 2) and so this will make this aspect challenging.
    3. Are the pie charts shown in fig 4 for individual tumours or are they an average across all the tumours subgrouped by the model of evolution? If the former – please label the pie charts with sample IDs; if the latter I don’t know what value this gives you given that every BC is really quite different in its genomic features.
  7. Section 2.6
    1. Line198 – it should be 3 ERBB2 amplifications and 1 mutation.
  8. Discussion – several statements don’t make so much sense and so please clarify them:
    1. 340-342 – I don’t think this is a critical observation, just that the recurrence is clonally related to the primary.
    2. Line 360 – what do you mean by “dysfuntion of the dna modification process”?
    3. 362-4 – is the point here that the pattern is different between primary vs recurrence and between a primary vs a metastasis? There is still considerable divergence between a primary and resulting metastasis.
    4. Take care around the discussion about HRD – as mentioned above it is not that striking a feature here and needs to be explored regarding the tumour cellularity of the samples.
    5. Can you make a comment, or think about, whether the time to relapse impacts the type of divergence in mutation spectra you see? Does a longer time drive more intratumour heterogeneity?
    6. Some discussion around the modelling of the types of tumour evolution might be useful. There are other studies to have done the same eg https://www.nature.com/articles/s41467-019-08593-4
    7. A section acknowledging the limitations of the study is important.

Round 2

Reviewer 2 Report

The authors have addressed a number of my previous comments and the paper has improved, thank you. Some remaining comments:

Section 2.2. lines 121-125. I don’t think you should group these 3 cases as the same. Yes 2 show HRD, but the 3rd germline BRCA2 case doesn’t show functional evidence of HRD. Better to say "2 relapse tumours harboured the HRD phenotype (HRD score >42). A further case exhibited biallelic germline alterations in BRCA without showing evidence of HRD (score >42)."

The new Fig 2 is better now the annotation is correct. Still perhaps don’t call it ‘germline biallelic alteration’ maybe ‘germline/somatic biallelic alteration’?

Line 147 –Saying 6/10 cases had higher CIN in the recurrence vs the primary is over stating the data and not robust. In 4 of these it’s a marginal difference. Robust evidence for 2/10. But this is fine.

Section 2.4 – clonal models – did you stratify the tumours into these 3 models or does the software do that for you? looking at the plots i dont see much difference between Fig 4a and 4b; both show the dominant clone(s) being eradicated by treatment presumably, and the emergence of new clones in the recurrence, driven by the acquisition of new mutations. so how are these different? Same with the images in Supp. you called the 3 with HRD ‘HRDEM’, but actually in only 2 of these is the evolution driven by HRD. The apparent HRD in case 7 is biallelic germline BRCA2 variants, but with no functional evidence of HRD by sequencing, so is this really HRD? So is the term ‘HRDEM’ misleading?

Line 228-9 – case BC01 didn’t have endocrine therapy according to Table S2 so this isn’t accurate.

Lines 264-5 “a minor clone survived to have a selective advantage in the relapsed tumor” this is the text used to describe the SEEM model, but is also exactly the same as what happens in the other models – isn’t it? They all exhibit this pattern whereby clones identified In the primary essentially die out, a minor clone survives and evolves in the recurrence. I think the key point is that this group of tumours exhibit more clonal heterogeneity in the primary and the recurrence compared to cases in the other models (and that maybe this is driven in part by APOBEC?)

Section 2.6. noting the BRCA2 germline case (BC07) here is ok, but you could also state that this patient might be given PARP or platinum based on these findings, but might NOT respond given the tumour wasnt HRD - this is as important a finding as others reported.

Noting case BC03 and the BRCA2 somatic mutation here is also ok, and you could mention that this recurrence had a notably higher HRD score compared to the matched primary but that it had an HRD score  <42 and so the benefit from PAPRi is not clear; but this is an interesting finding.

Discussion – I find it hard to pull out the key findings from this. sometimes things are a bit overstated and this distracts from the message. I think you are underpowered to say i) that primary to relapse divergence is higher than primary to metastasis; ii) you didn’t pick up signatures associated with specific treatments. You mention tumour purity in the limitations section, but not elsewhere, and so the statement on lines 466-67 doesn’t really make sense.

HRD is a major focus of the paper, yet the second paragraph of discussion doesn’t really pull out the key points around this. To me your key point is that in 4/10 cases there is a notable increase in HRD score in the recurrence vs the matched primary; in 2 of these the score is >42 and so perhaps these patients could be given HRD associated therapy (PARPi/platinum). However its unclear if any of these cases would respond given the scores are only just >42. In the other two cases PARPi/platinum is also a potential avenue, particularly in the case with somatic BRCA2 mutation. The case with BRCA2 germline mutation (BC07) might be considered for PARP/platinum however the tumour doesn’t exhibit HRD and so the efficacy of this approach is unclear.

Please also note this isn’t the first study to sequence primary tumours and locally relapsed tumours – Yates etal 10.1016/j.ccell.2017.07.005 . please include this article and thoroughly search the literature to ensure there are not others.
